# Leading consumption patterns of psychoactive substances in Colombia: A deep neural network-based clustering-oriented embedding approach

**Kevin Palomino** *, **Carmen R. Berdugo**, **Jorge I. Vélez**

Department of Industrial Engineering, Universidad del Norte, Barranquilla, Colombia

* krpalomino@uninorte.edu.co

## Abstract

The number of health-related incidents caused using illegal and legal psychoactive substances (PAS) has dramatically increased over two decades worldwide. In Colombia, the use of illicit substances has increased up to 10.3%, while the consumption alcohol and tobacco has increased to 84% and 12%, respectively. It is well-known that identifying drug consumption patterns in the general population is essential in reducing overall drug consumption. However, existing approaches do not incorporate Machine Learning and/or Deep Data Mining methods in combination with spatial techniques. To enhance our understanding of mental health issues related to PAS and assist in the development of national policies, here we present a novel Deep Neural Network-based Clustering-oriented Embedding Algorithm that incorporates an autoencoder and spatial techniques. The primary goal of our model is to identify general and spatial patterns of drug consumption and abuse, while also extracting relevant features from the input data and identifying clusters during the learning process. As a test case, we used the largest publicly available database of legal and illegal PAS consumption comprising 49,600 Colombian households. We estimated and geographically represented the prevalence of consumption and/or abuse of both PAS and non-PAS, while achieving statistically significant goodness-of-fit values. Our results indicate that region, sex, housing type, socioeconomic status, age, and variables related to household finances contribute to explaining the patterns of consumption and/or abuse of PAS. Additionally, we identified three distinct patterns of PAS consumption and/or abuse. At the spatial level, these patterns indicate concentrations of drug consumption in specific regions of the country, which are closely related to specific geographic locations and the prevailing social and environmental contexts. These findings can provide valuable insights to facilitate decision-making and develop national policies targeting specific groups given their cultural, geographic, and social conditions.

**Data Availability Statement:** The data used in the manuscript were obtained from a third party, the Archivo Nacional de Datos (ANDA), and are fully available and anonymized. The authors confirm

that others would be able to access these data in the same manner as themselves; and the authors did not have any special access privileges that others would not have. The data can be publicly retrieved from ANDA (https://microdatos.dane.gov.co/index.php/catalog/680/data-dictionary).

**Funding:** The author(s) received no specific funding for this work.

**Competing interests:** The authors have declared that no competing interests exist.

## 1. Introduction

Psychoactive substances (PAS) are chemical substances that change the function of the nervous system and cause alterations in people's perception, mood, consciousness, cognition, or behavior [1]. PAS can be grouped according to their chemical structure as synthetic cannabinoids, synthetic cathinones, phenethylamines, arylcyclohexylamines, tryptamines, indolalkylamines, new synthetic opioids, piperazines, ketamine, and designer benzodiazepines. They can also be grouped according to their origin as a natural origin or synthetic molecules [2–4].

The increased numbers of drug use among young people are drawing the attention of national governments [5]. Because the number of health-related incidents caused by using legal and illegal PAS worldwide has dramatically increased over the last two decades [2, 6], this phenomenon has become some of the largest burdens of disease [7, 8]. Drug use constitutes a high cost to society due to premature mortality, increased health expenditure, criminal justice (drug and micro-trafficking), social welfare costs, and other social consequences [9, 10].

Colombia is ranked as one of the largest drug producers in the world [11]. Unfortunately, the production and commercialization of drugs through drug- and micro-trafficking, constantly expands in locations with high levels of poverty and limited government presence [12]. Statistics and indicators of drug consumption, production, and distribution, as well as reports from the National Statistical System (DANE) of Colombia, highlight a dramatic increase in (i) drug production [13]; (ii) intern consumption of PAS at early ages; and (iii) the prevalence in use and/or abuse of drugs have dramatically increased over the last 20 years [14, 15]. Furthermore, the country also has the highest prevalence of drug use among school students in recent years compared to other Latin American countries [16]. Thus, there is an urgent need to develop effective interventions to prevent the use and/or abuse of PAS.

The first step towards reducing this consumption is to identify drug consumption patterns in the general population [17]. Several studies have identified patterns associated with drug use and consumption [18, 19]. According to the Center for Disease Control and Prevention, individuals who do not have their own homes and live in rented accommodations are more likely to use drugs [20]. Other research studies suggest that neighborhood contextual characteristics may increase the risk of substance abuse [21–26]. Additionally, population density may also influence substance use and overdose risk through a higher level of socialization in densely populated urban areas [27–29]. Other authors have identified that anxiety, sleep disorders, suicide, depression, and other mental illnesses are risk factors for the consumption and abuse of PAS [30, 31]. Furthermore, early marijuana use has been shown to increase the risk of consuming other PAS [32]. Furthermore, people involved in sports and artistic activities perceive drugs as enhancers element for improving their performance [33].

In Colombia, drug consumption patterns and risk factors have also been identified. For instance, Kalyanam et al. [34] analyzed the social impact of basuco and inhalant use among street youths. Narvaez-Chicaiza [35] assessed the social factors that lead to the adoption of harm reduction policies and how these factors influence treatments for substance abuse disorders. Additionally, Restrepo-Escobar & Cardona [36] demonstrated that university students with low satisfaction in their studies tend to be heavy users of alcohol, tobacco, and marijuana. However, these approaches do not consider the use of Machine Learning (ML) and/or Deep Data Mining techniques in combination with spatial models to analyze drug consumption data from the general population. To our knowledge, we have not found any robust models integrating ML and spatial models to identify drug consumption patterns using publicly available Colombian databases.

Although several techniques for analyzing drug consumption patterns are currently available (i.e., ML, Bayesian, spatial, traditional multivariate, or univariate statistical models, or, in

some cases, a combination of these), new trends in pattern identification and analysis techniques focus on hybrid and ensemble models [37]. Currently, the most widely used ML techniques are Support Vector Machines (SVMs), Random Forest (RF), and Natural Language Processing (NLP) [38–42]. Among Bayesian models, Bayesian meta-regression (DisMod-MR), Bayesian hierarchical models, and Markov Chain Monte Carlo are the most attractive methods [43–45]. Regarding spatial models, Spatial Distributions, Spatial Regression Models, Spatial Scan Statistics, Variograms, and Social Mapping are the most frequently used techniques [46–49]. On the other hand, logistic regression, confirmatory factor analysis, and correlational analysis are the most employed traditional statistical models to identify drug-associated patterns [50–53].

Fraley and Raftery [54] suggest separating clustering approaches into hierarchical and partitioning techniques. Partitioning techniques are divided into density-, model-, and grid-based methods, the most popular of which are *K*-means, PAM, CLARA, DBSCAN and CLIQUE. On the other hand, hierarchical techniques are divided into agglomerative and divisive methods. Of these, the best-known methods are BRICH, CURE, ROCK, and CHAMELEON (see [55] for further reading). Although these techniques have been shown to perform well when relevant features are removed *a priori*, it is well-known that in clustering algorithms, irrelevant and redundant features in the data may degrade the quality of clusters and lead to high computational cost. Therefore, removing such features may alleviate these issues. Thus, we focus on identifying patterns of PAS consumption using an ensemble model integrating an autoencoder with both a clustering algorithm and a spatial model. As part of our approach, we used the most recent and representative works for data clustering, and different dimensionality reduction and feature selection methodologies proposed in the literature.

Feature selection approaches in clustering can be split into filter, wrapper, embedded, and hybrid approaches [37]. While wrappers depend on the clustering algorithms to evaluate the clustering quality of a selected feature subset, filters are independent of the clustering algorithm. Embedded approaches also work with a clustering algorithm and, unlike wrappers, incorporate knowledge about the clustering structure. Another type of method is hybrid approaches, which combine filter and wrapper approaches into a single strategy. However, studies on embedded and hybrid feature selection approaches in clustering are limited [37]. Other feature learning-based approaches using Deep Neural Networks have been shown to work well for linear and nonlinear models [56]. For instance, Xie et al. in (2016) [57] propose to work on feature extraction and clustering using pre-trained Auto-Encoders simultaneously. However, these are mainly used to work and process images. In general, deep clustering models use Auto-Encoders since they can learn input features without labels on the data; performance measures show that this approach is reliable for different data types [58]. Thus, deep clustering methods have become a growing field of research for feature selection [58]. In this regard, the use of convolutional networks in autoencoders and the application of feature selection for clustering are open questions that have not been fully addressed yet, especially when dealing with data from different statistical distributions [37].

Here, we propose a Deep Neural Network-based Clustering-oriented Embedding Algorithm that allows us to (*i*) identify consumption patterns of PAS; and (*ii*) build an ensemble algorithm integrating an autoencoder with a clustering algorithm and a spatial model to deal with the feature space and cluster memberships. Our approach is based on the model proposed by Xie et al. [57] and B. Li et al. [56], and expands their work by creating an autoencoder from a convolutional network to represent high-order interactions in the data accurately and simultaneously incorporate a spatial analysis to describe drug consumption patterns properly. Our main hypothesis is that incorporating these two critical elements in our proposal will help to

identify and better understand drug consumption patterns and support national policy development processes.

## 2. Materials and methods

### 2.1 Study area

Located in South America, the Republic of Colombia is a diverse country with a population of over 50 million people distributed over a territory of 440,831 square miles [59], encompassing jungles, highlands, grasslands, deserts, coasts, and islands, distributed in six regions and 32 departments (states) [60] (See S1 Fig). It is worth noting that, unfortunately, Colombia has been a major producer of illegal drugs for a long time, which has had a significant impact on drug consumption and abuse.

According to the United Nations Office on Drugs and Crime, Colombia is the first cocaine-producing country and the eighth country with the highest production of cannabis [61]. In addition, the Colombian Drug Observatory indicates that the use of illicit substances in the territory has increased to 10.3%, with men between the ages of 18 and 24 being the heaviest consumers of these types of drugs. Reports also indicate that consumption of licit substances such as alcohol and tobacco has recently increased dramatically [62].

### 2.2 Data sources

We used two databases to identify drug consumption patterns in Colombia. The first database was retrieved from the 2019 National Survey of Psychoactive Substance Consumption in the General Population (DANE-DIMPE-ENCSPA-2019; URL: https://microdatos.dane.gov.co/index.php/catalog/680/data-dictionary) conducted by the National Statistical System (DANE) of Colombia [63]. This survey includes observations of 49,600 households, where information on housing, location, general characteristics of individuals, consumption of legal and illegal PAS, and implemented treatments is registered. The second database comes from the Colombian Drug Observatory and contains information on the production of PAS per area during 2019. All these databases are fully available and completely anonymized. In this study, we used departments (states) as georeferenced areas using polygons (i.e., a shapefile) as implemented in ArcGIS Hub [64]. Thus, an ethics statement approved by an ethics committee is not required since we are using public information without the identification or individual information of the people involved.

### 2.3 Convolutional Auto-Encoder-Deep Embedded Clustering algorithm

Fig 1 presents the proposed Convolutional Auto-Encoder- Deep Embedded Clustering (CAE-DEC) framework based on the implementation presented by Xie et al. [57]. However, unlike the Xie et al. model, our structure is developed by applying convolutional layers for the deep autoencoder (DA) architecture instead of a linear one to represent high-order interactions in the data. In addition, a spectral clustering-based centroid estimation is proposed to achieve an improved initial centroid calculation. We chose the CAE-DEC framework based on its ability to reduce both the number of model parameters and the dimensionality, while creating clusters simultaneously.

In our approach, an encoder structure is first applied to map the input vector into a lower feature space, called latent feature space (LFS). Then, the LFS is independently passed through a decoder structure and a clustering layer to achieve an efficient clustering framework. The encoder-decoder combination (DA) attempts to extract a LFS preserving the relevant information from the original input data. On the other hand, the clustering layer seeks to execute an

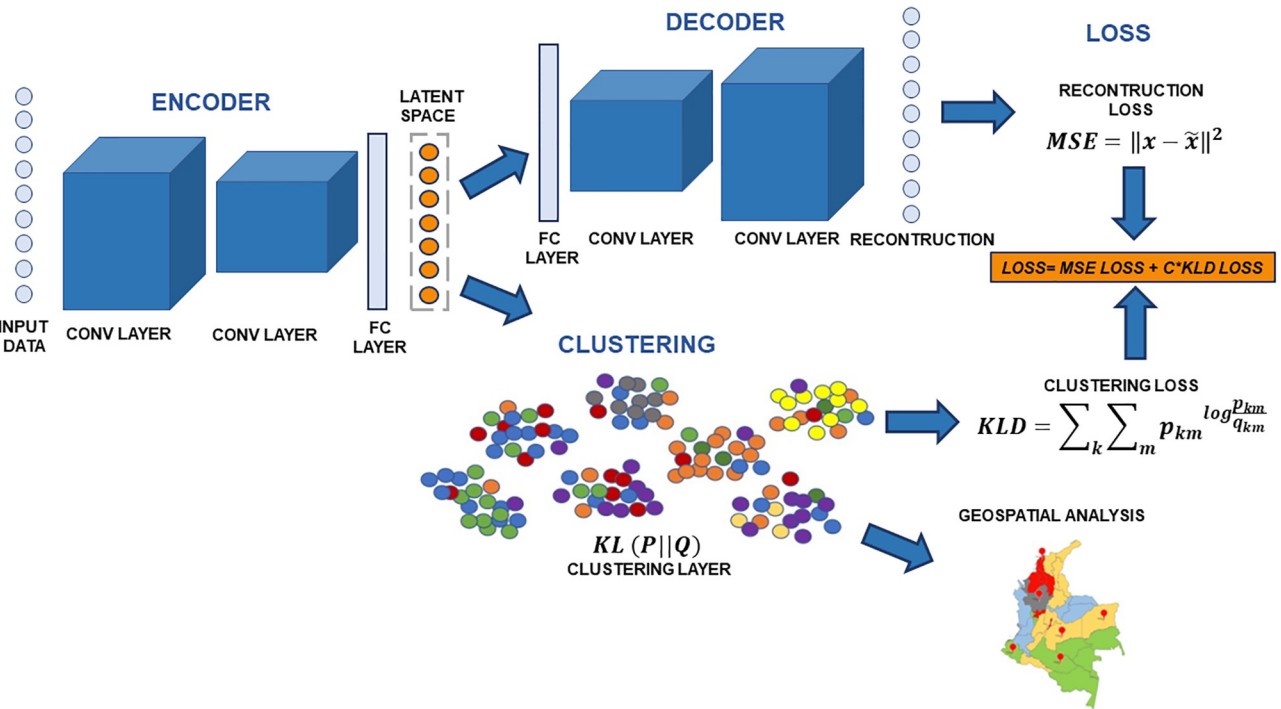

**Fig 1. Architecture of the proposed CAE-DEC model.** Republished from [64] under a CC BY license, with permission from [ArcGIS Hub], original copyright [2016].

improved clustering assignment by minimizing the divergence between a target distribution and a centroid-based probability distribution.

In the last stage of the framework, a spatial analysis was performed using the feature space generated from the autoencoder as input. Here, the spatial data exploration is initially performed using Global Spatial Autocorrelation to determine to which level the similarity between observations in a dataset relates to the similarity of the locations of such observations [65]. To assess GSA, the Moran's I [66], Geary's *C* [67], and Getis and Ord's *G* [68] statistics are estimated. We also measure the Local Spatial Autocorrelation, which focuses on the relationships between each observation and its surroundings, rather than providing a single-number summary of these relationships across the map [69]. This is estimated based on the ability to determine whether spatial autocorrelation is present in a geographically referenced data set. Finally, we perform regionalization, which corresponds to a special kind of clustering where the objective is to group similar observations based on their statistical attributes and spatial location [70]. In this sense, regionalization embeds the same logic as standard clustering techniques while applying a series of geographical constraints [71]. This framework was built using the TensorFlow (https://www.tensorflow.org/) and PyTorch (https://pytorch.org/) libraries in Python version 3.11 [72].

## 2.4 Convolutional Auto-Encoder (CAE)

The DA is a deep neural network architecture capable of learning unsupervised representations of an input data set. Typically, DA networks are used for dimensionality reduction or denoising tasks. The structure of a DA is based on two deep networks: a network to transform the original input data into a latent feature space, and a network trained to reconstruct the

original input data using the extracted latent space as input. The first network, used to extract the latent space, is called the encoder, while the second is called the decoder. Rather than using fully connected layers, the implemented DA architecture incorporates convolutional (CONV) layers and fully connected (FC) layers for LFS extraction and reconstruction (Fig 1). Integrating convolutional layers in a DA is also called CAE [73]. Compared to a DA, which is built with only fully connected layers, the CAE structure can reduce the number of parameters compared to a DA [74].

**2.4.1 Convolutional layer.** The proposed CAE structure is designed using four convolutional layers, two CONV layers during the encoder stage, two CONV layers during the decoder stage, and two fully connected layers (Fig 1). The convolution operation can be denoted as:

$$z_{i,j,k}^{l} = W_{k}^{l^{T}} X_{i,j}^{l} + b_{k}^{l} \tag{1}$$

where $z_{i,j,k}^{l}$ is the value of each feature at the $(i, j)$ location in the $k$-th feature map of the $l$-th layer, $W_{k}^{l}$ and $b_{k}^{l}$ represent the weight and bias of the $k$-th filter of the $l$-th layer, and $X_{i,j}^{l}$ denotes the input value at location $(i, j)$ of the $l$-th layer. For non-linear mapping, an activation function $g(.)$ is applied over the convolutional feature $z_{i,j,k}^{l}$ as follows:

$$a_{i,j,k}^{l} = g(z_{i,j,k}^{l}) \tag{2}$$

where $a_{i,j,k}^{l}$ is the activation value resulting from applying the activation function $g(.)$. The Rectified Linear Unit (ReLU) function is set as the activation function on each CONV, except in the final decoder CONV layer where a sigmoidal activation is applied.

## 2.5 Clustering layer

The clustering layer is inspired by Xie et al. [57]. Initially, a soft assignment is computed between the latent space, also known as embedded space, and the cluster centroids. Then, update steps are repeated to define the final cluster centroids and embedded space. The Kullback–Leibler (KL) divergence is used as loss function during the optimization procedure. The objective is to minimize de KL divergence between a soft clustering distribution $Q$ and an auxiliar target distribution $P$. The KL loss is calculated as:

$$L_c = KL(P||Q) = \sum_k \sum_m p_{km} log\left(\frac{p_{km}}{q_{km}}\right) \tag{3}$$

where $L_c$ is the clustering loss. To measure the similarity between embedded point $z_k$ and the cluster centroid $c_m$, the $t$ Student's distribution is used as a kernel:

$$q_{km} = \frac{\left(1 + ||z_k - c_m||^2/\alpha\right)^{-\frac{\alpha+1}{2}}}{\sum_i \left(1 + ||z_k - c_i||^2/\alpha\right)^{-\frac{\alpha+1}{2}}} \tag{4}$$

with $\alpha$ the degrees of freedom of the $t$ Student's distribution and $q_{km}$ is a soft clustering assignment distribution of each embedded point (i.e., probability of assigning point $k$ to cluster $m$). As in Xie et al., when setting $\alpha = 1$ the similarity function $q_{km}$ can be calculated as:

$$q_{km} = \frac{\left(1 + ||z_k - c_m||^2\right)^{-1}}{\sum_i \left(1 + ||z_k - c_i||^2\right)^{-1}} \tag{5}$$

To compute the target distribution $p_{km}$, the second power of $q_{km}$ is calculated, and a cluster normalization is applied as follows:

$$p_{km} = \frac{q_{km}^2 / \sum_k q_{km}}{\sum_i \left( q_{ki}^2 / \sum_k q_{ki} \right)} \qquad (6)$$

Then, by minimizing the divergence between $P$ and $Q$, the embedding learning is achieved through highly confident assignments.

**2.5.1 Center initialization.** As previously mentioned, the cluster centroids are initialized using a spectral clustering-based approach. The spectral clustering allows flexible distance metrics and provides better cluster estimations than $K$-means [57]. However, most spectral clustering algorithms have high computational requirements. To overcome these computational requirements, random samples are taken to estimate the cluster centroids. As spectral clustering does not estimate any centroid during the learning process, once the clusters are defined, the mean of each cluster is used as the centroid estimator.

## 2.6 The CAE-DEC model

Initially, the input data is normalized within the interval [0, 1]. This normalization allows the network to use the most advanced learning rate and avoid the vanishing gradient problems, as well as alleviate overfitting. Further, to achieve a better learning process, the last CONV layer in the decoder structure is activated by a sigmoid activation function. Then, two training steps will be executed during the CAE-DEC learning process. Firstly, a CAE model will be trained to minimize the reconstruction loss $L_r$ computed as

$$L_r = ||x - \tilde{x}||^2 \qquad (7)$$

where $x$ is the normalized input and $\tilde{x}$ is the reconstructed output. This pretrained CAE model is then used as the DA structure in the CAE-DEC model.

In the second step, the CAE-DEC model is trained to simultaneously minimize reconstruction loss and clustering loss. The total loss during this training step will be set as

$$L_t = L_r + C \cdot L_c \qquad (8)$$

where $L_r$ is the CAE-DEC reconstruction loss, $L_c$ is the CAE-DEC clustering loss, and $C$ is a coefficient to control the loss balance. The training process is shown in Table 1. The goal is to obtain a latent space that minimizes the total loss. Finally, the label of each embedded point is established as

$$Label_j = \arg max_m q_{jm} \qquad (9)$$

where $q_{jm}$ is the probability that point $j$ belongs to a specific cluster center $m$. On the other hand, the maximum number of iterations Mint and the target distribution P update condition P_change was chosen based on multiple experiments. The final Mint and P_change values were 3000 and 5, respectively. This final P_change improves stability during the training process.

## 2.7 Framework evaluation

We trained the CAE-DEC method using data retrieved from the National Survey of Psychoactive Substance Consumption (DANE-DIMPE-ENCSPA-2019), which contains 49,600 observations. A second database with PAS production figures, was used in the spatial analysis stage to correlate the PSA consumption and production. In order to evaluate the framework, we

**Table 1. Pseudo code for the CAE-DEC training process.**

| Pseudo code: The CAE-DEC training process |
|---|
| Input data: Number of clusters n; Normalized input data x; Maximum number of iterations Mint; Balance coefficient C; Pretrained CAE; Stop condition Stop; Target distribution P update condition P_change. |
| Training process: |
| 1. Generate an initial latent space (Z) through the pre-trained CAE |
| 2. Run spectral clustering with Z to generate the initial cluster centers (C) |
| 3. Initialize the CAE-DEC model with the pretrained CAE. |
| 4. Calculate soft assignment distribution Q and target distribution P based on $Z$ and $C$ |
| for epoch<Minter do: |
| if epoch%P_change = = 0 then: |
| Calculate soft assignment distribution Q and target distribution P based on $Z$ and $C$ |
| end if |
| Feed the CAE-DEC with the normalized input data x |
| Calculate the reconstruction loss and the clustering loss |
| Update CAE-DEC parameters. Weight, Bias, and Centers. |
| if Stop = = True then: |
| Break |
| end if |
| end for |
| Obtain the label for each data point from the las optimized Q. |
| Output: Latent space, labels |

compared our CAE-DEC approach with other approaches, including CAE, and Principal Component Analysis integrated with clustering (PCA-DEC). For evaluation and comparison purposes, we use the Calinski-Harabasz [75], Davies-Bouldin [76], and Silhouette [77] index as intrinsic clustering metrics. In addition, we used the $\chi^2$ statistic to investigate potential associations and differences among the patterns (clusters) identified using our approach.

## 3. Results

### 3.1 Model comparison for identifying drug consumption patterns

Fig 2 depicts the LFS resulting after applying the CAE and CAE-DEC models to the data. Among all individuals, we identified three different clusters; 14935 (30.19%) individuals belong to cluster 0, 11528 (23.30%) individuals belong to cluster 1, and 23005 (46.50%) individuals belong to cluster 2. Interestingly, the LFS generated with the CAE-DEC has more defined clusters than the CAE model. Although the CAE model seeks to extract a LFS that preserves the essential characteristics of the input data, our proposed CAE-DEC model not only preserves these important characteristics but, at the same time, also forces the encoder structure to generate representative clusters while extracting the new feature space.

On the other hand, the reconstruction loss obtained through the CAE model is higher than that of the CAE-DEC model. This result may be related to the fact that the CAE-DEC model used the pre-trained CAE model during its construction. It should be noted that the CAE model alone cannot determine the labels of each point or define clusters in the data. Thus, clusters in Fig 2 were obtained through spectral clustering and were the bases for initializing the centroids in the CAE-DEC model.

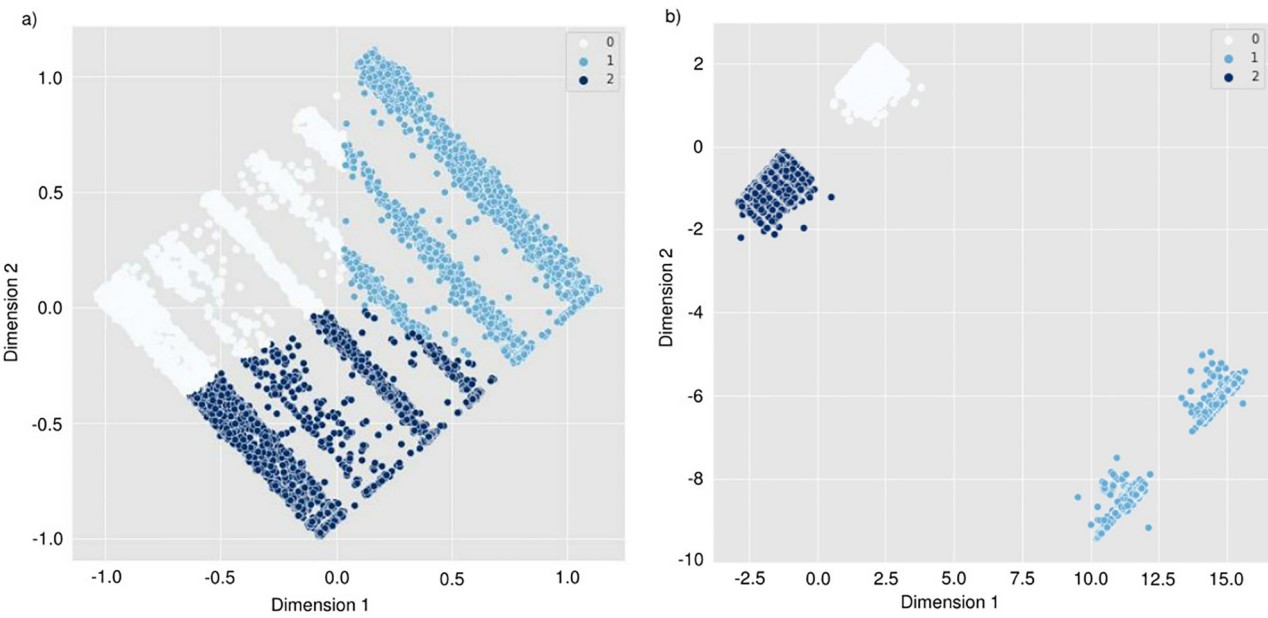

**Fig 2. Derived Latent Feature Space based on the (a) CAE and (b) CAE-DEC models.**

## 3.2 Identification of clusters of psychoactive drugs consumption

Here we analyse the patterns in each cluster obtained using the CAE-DEC model. We defined a priority dummy variable $Y_{ij}$ quantifying whether the $i$th person in household $j$th has consumed PAS; $Y_{ij} = 1$ when an individual has never consumed PAS and $Y_{ij} = 2$ otherwise. Out of the 49468 individuals in the sample, only 5514 (11.15%) consume PAS. Fig 3a and 3b depict,

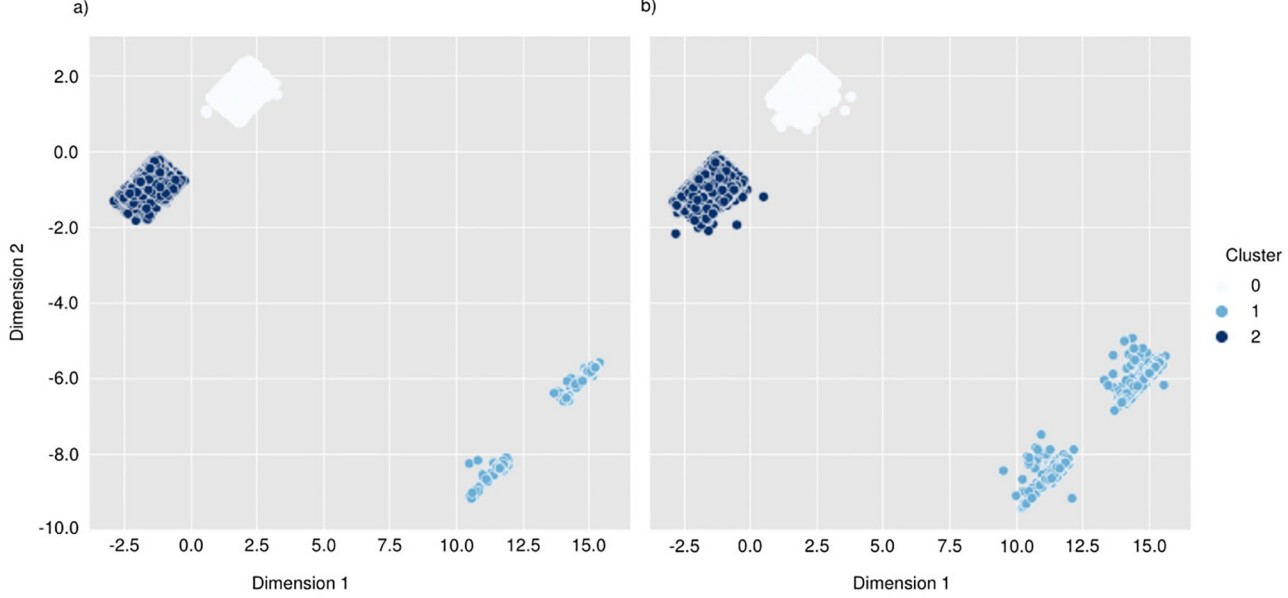

**Fig 3. Resulting clusters for individuals (a) consuming and (b) not consuming psychoactive substances based on the CAE-DEC model.**

respectively, the derived cluster structure for individuals consuming PAS and those who reported not consuming, derived from the CAE-DEC model. Our results indicate that individuals in clusters 0 and 2 are more likely to consume some PAS (Fig 3a), while most individuals in cluster 1 do not (Table 2). In particular, 1726 (11.56%) individuals in cluster 0, 392 (3.4%) individuals in cluster 1, and 3396 (14.76%) individuals in cluster 2 have used PAS (Table 2). A $\chi^2$-based test of independence reveals that the region where individuals are located, age (years), the type of household they live in, their socioeconomic status (SES), and whether they contribute to the household finances are statistically significantly associated with the cluster they belong to (Table 2).

Table 3 shows the adjusted residuals for our model. According to our results, the Central-Eastern region significantly contributes to the Region variable. In this region, the observed value is higher than the expected value in cluster 2, while the observed value is lower than the expected value for cluster 0. Although to a lesser extent, the Llanos Orientales region also significantly contributes the $\chi^2$ statistic. Indeed, this region shows fewer observed individuals than the expected number of individuals in cluster 2 and a higher number observed than expected individuals in clusters 0 and 1 (Table 3).

On the other hand, Gender has a higher-than-expected value of males in clusters 0 and 2, while it is lower in cluster 1. For females, the opposite occurs in cluster 1, and lower values are observed in clusters 0 and 2. Similarly, Housing Type has a higher-than-expected value of individuals living at houses in cluster 1 and a lower-than-expected in cluster 2. Conversely, cluster 2 has more individuals living in apartments, and cluster 1 has the lowest (Table 3).

**Table 2. Distribution of demographic and social variables across clusters.**

| Variables | | Cluster 0 ($n$ = 14935) | Cluster 1 ($n$ = 11528) | Cluster 2 ($n$ = 23005) | $\chi^2$ | df | P-value |
|---|---|---|---|---|---|---|---|
| Region | Caribbean | 3004 | 2991 | 4075 | 1472.9 | 10 | < .0001 |
| | Central-Eastern | 2526 | 2299 | 6175 | | | |
| | Central-Southern | 1843 | 1299 | 1702 | | | |
| | Eje Cafetero–Antioquia | 3902 | 2354 | 6371 | | | |
| | Llanos Orientales | 1687 | 1305 | 1496 | | | |
| | Pacific | 1973 | 1280 | 3186 | | | |
| Gender | Male | 6606 | 3927 | 10233 | 386.72 | 2 | < .0001 |
| | Female | 8329 | 7601 | 12772 | | | |
| Housing type | House | 8250 | 6538 | 11834 | 114.18 | 6 | < .0001 |
| | Apartment | 6275 | 4724 | 10595 | | | |
| | Room | 395 | 256 | 543 | | | |
| | Indigenous dwelling | 15 | 10 | 33 | | | |
| Socioeconomic status | 1 | 3889 | 3626 | 6945 | 843.92 | 10 | < .0001 |
| | 2 | 4627 | 3902 | 8851 | | | |
| | 3 | 4284 | 2867 | 5683 | | | |
| | 4 | 1341 | 718 | 979 | | | |
| | 5 | 510 | 256 | 362 | | | |
| | 6 | 284 | 159 | 185 | | | |
| Age (years) | (0, 20] | 2105 | 1576 | 2998 | 345.87 | 4 | < .0001 |
| | (20, 40] | 6814 | 4329 | 10889 | | | |
| | (40, 68] | 6016 | 5623 | 9118 | | | |
| Contribute to the household finances | Yes | 10128 | 7473 | 16041 | 85.24 | 2 | < .0001 |
| | No | 4807 | 4055 | 6964 | | | |

*df*: Degrees of freedom.

**Table 3. Adjusted residuals comparing the observed and expected frequencies based on the cluster analysis.**

| Variables | | Cluster 0 ($n$ = 14935) | Cluster 1 ($n$ = 11528) | Cluster 2 ($n$ = 23005) |
|---|---|---|---|---|
| Region | Caribbean | -0.88 | 17.02 | -13.61 |
| | Central-Eastern | -18.72 | -6.76 | 22.97 |
| | Central-Southern | 12.54 | 6.09 | -16.7 |
| | Eje Cafetero—Antioquia | 2.02 | -14.36 | 10.31 |
| | Llanos Orientales | 11.32 | 9.59 | -18.55 |
| | Pacific | 0.84 | -6.97 | 5.13 |
| Gender | Male | 6.68 | -19.66 | 10.52 |
| | Female | -6.68 | 19.66 | -10.52 |
| Housing type | House | 4.17 | 7.13 | -9.88 |
| | Apartment | -4.83 | -6.61 | 10.05 |
| | Room | 2.2 | -1.54 | -0.72 |
| | Indigenous dwelling | -0.72 | -1.09 | 1.59 |
| Socioeconomic status | 1 | -10.26 | 5.99 | 4.37 |
| | 2 | -12.72 | -3.3 | 14.51 |
| | 3 | 9.14 | -3 | -5.87 |
| | 4 | 17.29 | 0.44 | -16.29 |
| | 5 | 11.12 | -0.49 | -9.82 |
| | 6 | 8.26 | 1.2 | -8.62 |
| Age (years) | (0, 20] | 2.54 | 0.61 | -2.85 |
| | (20, 40] | 3.2 | -17.23 | 11.66 |
| | (40, 68] | -4.98 | 16.93 | -9.77 |
| Contribute to the household finances | Yes | -0.61 | -8.37 | 7.65 |
| | No | 0.61 | 8.37 | -7.65 |

Regarding SES, a higher-than-expected number of individuals in strata 3, 4, 5, and 6 in cluster 0 were found (Table 3). We also observed a lower-than-expected number of individuals in strata 3, 4, 5, and 6 in cluster 2 and a higher-than-expected number in strata 1, 4 and 6 in cluster 1 (Table 3). Moreover, the age variable shows a higher-than-expected observed value for the (0,20] range in cluster 0. For ages between (20,40] years, cluster 2 has a higher-than-expected number of individuals. Conversely, there is a lower number of individuals in cluster 1. Finally, the household economy variable results show that cluster 2 has a higher-than-expected value of individuals contributing to the household finances, and cluster 1 has a lower-than-expected value of individuals not contributing to it. Comparison of Calinski-Harabasz, Davies-Bouldin, and silhouette metrics between a principal component analysis (PCA)-based deep autoencoder (PCA-DEC) and our proposed CAE-DEC model indicates the superiority of the latter (S2 Table).

## 3.3 Spatial analysis of psychoactive drugs consumption

Different alternative classification algorithms were used to determine the number of choropleth class limits (i.e., Equal Intervals, Quantiles, Maximum Breaks, Box plot, Head-Tail Breaks, Jenks-Caspall, Fisher-Jenks, and Max-p) and compared using the absolute deviation around class medians optimization criterion (Fig 4). According to our results, the Fisher-Jenks classifier performed better and hence was selected.

Following the same exploratory spatial analysis, we constructed a choropleth with the percentage of PAS use for each of the 32 Colombian departments (Fig 5a). We found that the

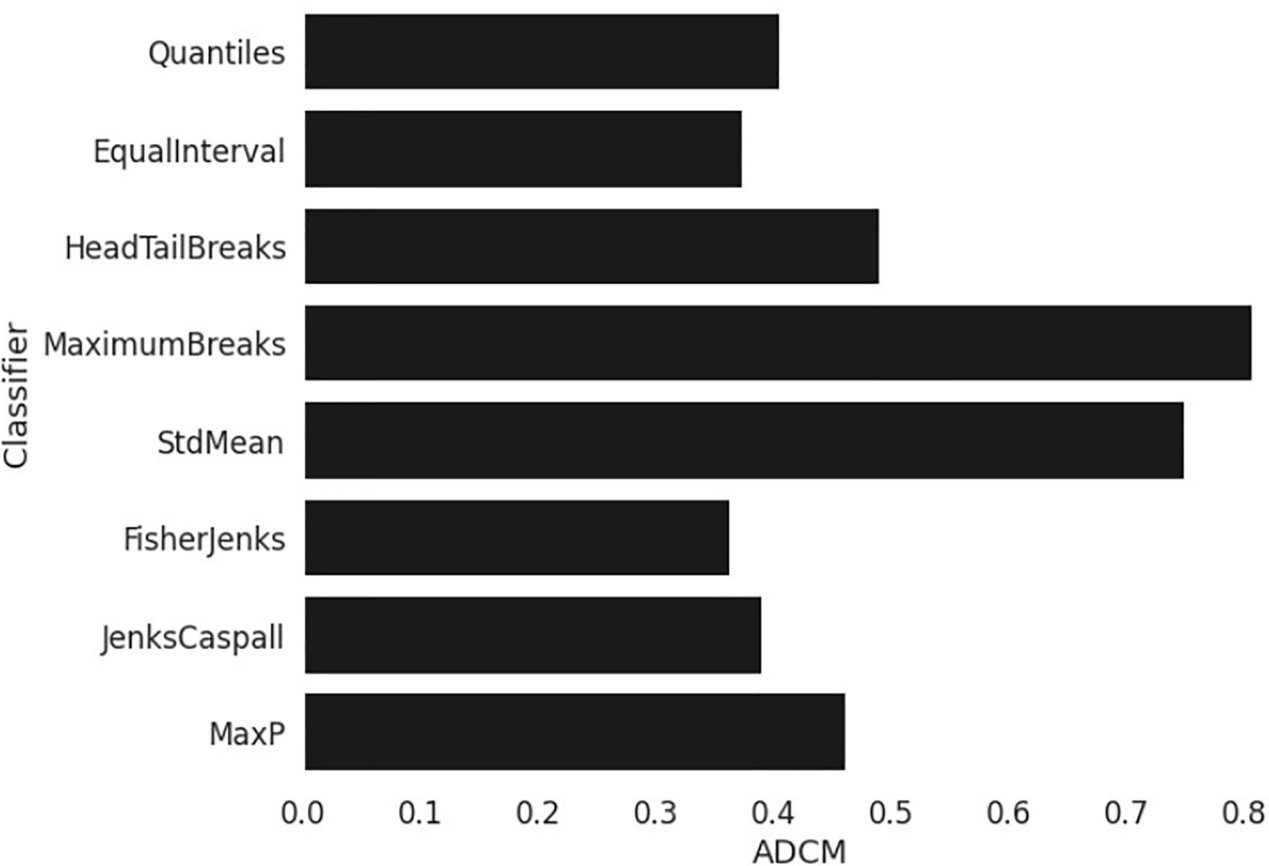

**Fig 4. Absolute deviation around class medians (ADCM) statistic criterion for different alternative classifiers.** Here, lower is better.

departments of Arauca, Vichada, Caquetá, Chocó, Magdalena, Cesar, Bolivar, Sucre, Cordoba, and Norte de Santander have low percentages of drug use. However, some of these departments are major drug producers (i.e., Cordoba and Guaviare), according to data from the Drug Observatory of Colombia [78]. Similarly, Putumayo is the department with the highest proportions of PAS use (Fig 5a).

The global Moran's I results show the presence of a statistically significant positive global spatial autocorrelation (I = 0.2005, $P{<}0.01$). Thus, the null hypothesis that the map is random (i.e., that the map shows more spatial patterns than we would expect if the values had been randomly assigned to a location) is rejected. In addition, other global indices such as Geary's $C$ ($C$ = 0.693, $P$ = 0.003) and Getis and Ord's $G$ ($G$ = 0.800, $P$ = 0.049) confirm the presence of statistically significant global spatial autocorrelation.

To further explore the relationships between each observation and its environment, the Local Indicators of Spatial Association (LISA) were estimated (more information on LISA statistics is provided in S3 Fig). Fig 5b depicts the Moran diagram, indicating each quadrant's positive (or negative) association. Specifically, the high-high (HH) and low-low (LL) quadrants indicate a positive association between high and low drug use. On the other hand, the low-high (LH) and high-low (HL) quadrants indicate negative associations with drug use (Fig 5b). Following our results, we found that departments such as Nariño and Cauca belong to the HH cluster. In contrast, la Guajira, Atlántico, Magdalena, Cesar, Norte de Santander, Sucre, and

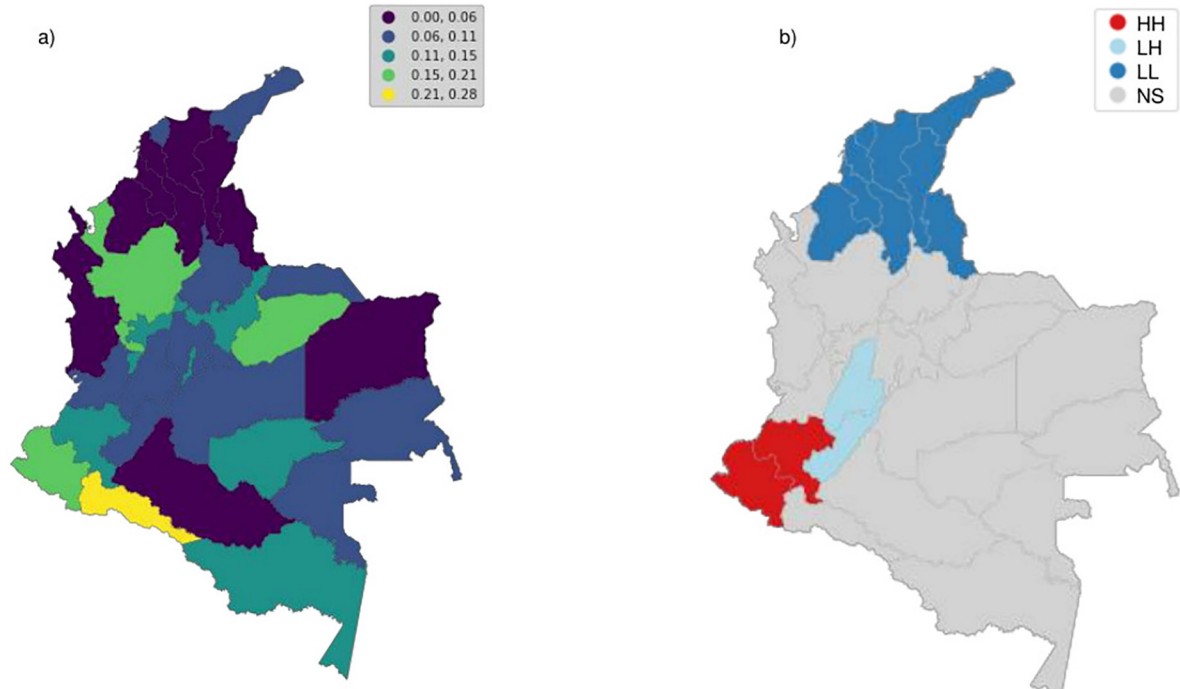

**Fig 5. (a) Consumption percentage of psychoactive substances according to the CAE-DEC model; (b) Moran's I statistic; (c) Moran's cluster map.** Here, HH, LH, LL and ns represent high-high, low-high, low-low, and not statistically significant quadrants, respectively. This clustering pattern leads to a statistically significant Moran's I statistic of 0.2 (*P*-value <0.01). Architecture of the proposed CAE-DEC model. Republished from [64] under a CC BY license, with permission from [ArcGIS Hub], original copyright [2016].

Cordoba belong to the LL. This clustering pattern leads to a statistically significant Moran's I statistic (*P*-value <0.01). Thus, a little over 39.4% of the departments are considered, by this analysis, to be part of a spatial cluster (i.e., statistically significant with a *P*-value <5%). We also identified that, among legal drugs, alcohol and tobacco are the most frequently consumed in the national territory (Fig 6a). At the same time, marijuana, followed by non-prescription tranquilizers and Yagé, and a slight consumption of opioids and Poppers, are the most frequently consumed illegal drugs (Fig 6b).

Regarding legal drugs, alcohol has the highest consumption rates in Bogotá, Cundinamarca, and Chocó (Fig 7). However, there is moderately high use in Vaupés, Nariño, Bolívar, Magdalena, La Guajira, and Atlántico. As for energy drinks, consumption is the highest in Casanare and Guaviare and has slightly high uses in Boyacá, Nariño, Risaralda, and Arauca. On the other hand, tobacco has the highest consumption in Cundinamarca but has moderately high uses in Bogotá, Boyacá, Nariño, Casanare, Tolima, Quindío, Risaralda, Guainía, Caldas, and Vaupés. It should be mentioned that the use of these drugs is also present across the country but with a lower incidence (Fig 7).

Concerning illegal drugs, non-prescription tranquilizers and stimulants are most prevalent in Casanare (Fig 8). However, the consumption of tranquilizers is slightly higher in Nariño, while inhalants have the highest consumption in Quindío, followed by Cauca, Caldas, and Nariño. Methylene Chloride has the highest consumption in Cauca and a high consumption in Quindío and Nariño; Antioquia, followed by Caldas and Risaralda, shows the highest consumption of popper. On the contrary, marijuana has its highest consumption in Risaralda and

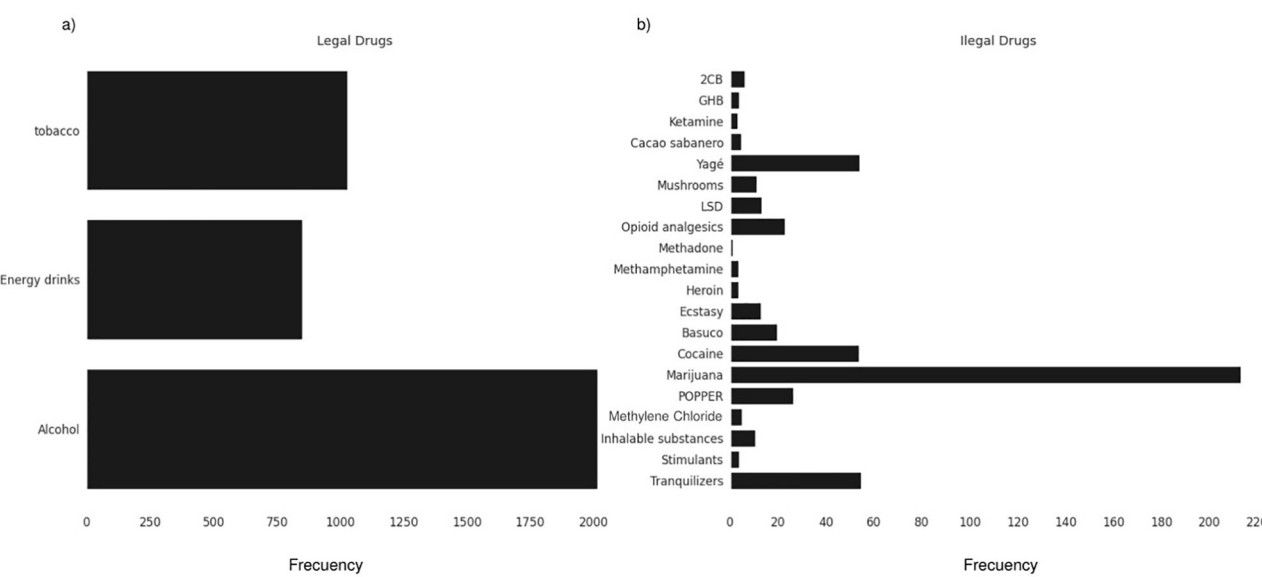

**Fig 6. Frequency of consumption of (a) legal and (b) illegal drugs in Colombia.**

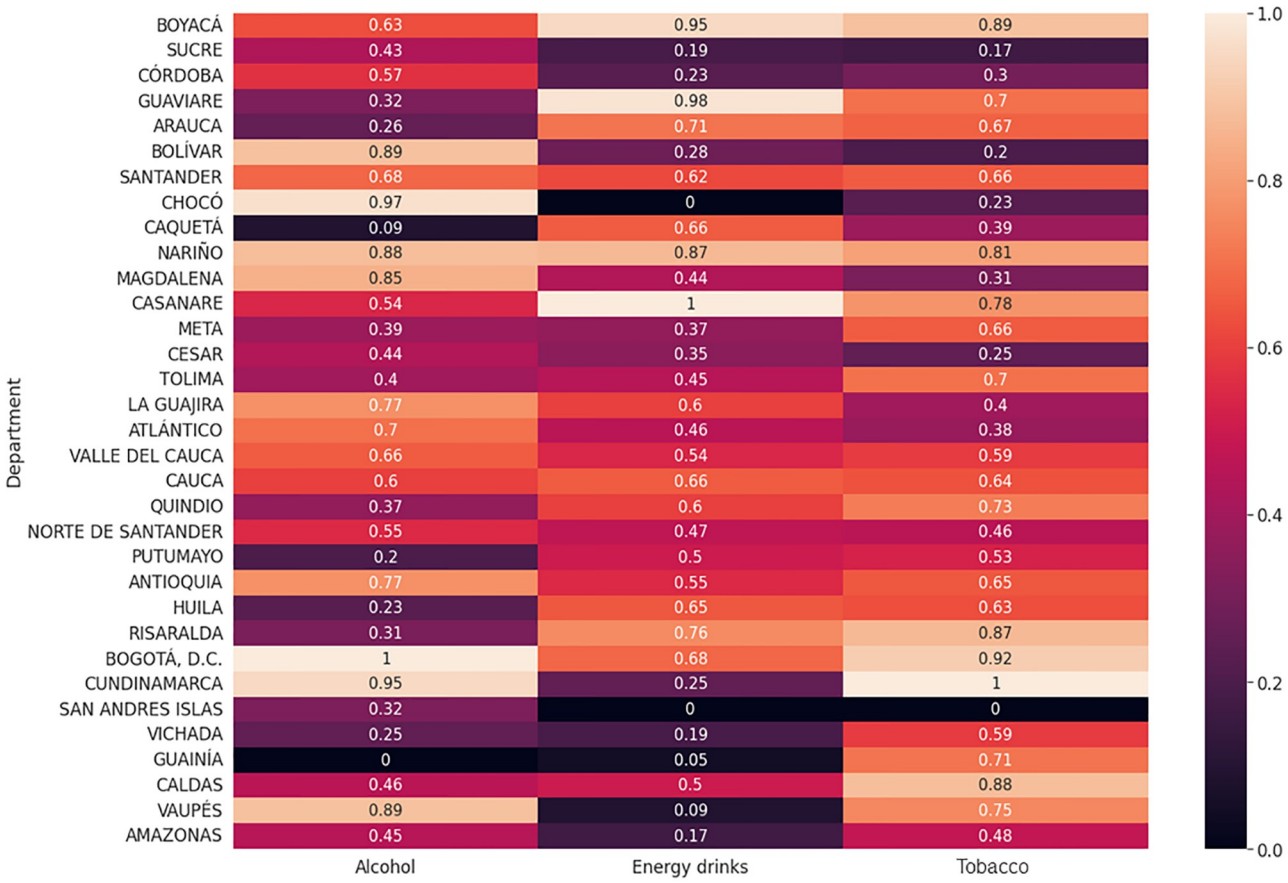

**Fig 7. Consumption of illegal drugs by department.** For interpretation purposes, number represents values scaled on a range of 0 to 1. For instance, Bogotá D.C. has the highest LSD consumption and Putumayo has the lowest.

moderately high consumption in Caldas, Bogotá, Antioquia, and Quindío. As for cocaine, its consumption is the highest in Risaralda and moderately high in Antioquia (Fig 8).

On the other hand, basuco (i.e., cocaine paste) has the highest consumption rate in Guaviare, and critical consumption in Nariño, Cauca, Quindío, Antioquia, and Amazonas; ecstasy has its highest consumption in Risaralda, followed by Bogotá and Caldas; heroin consumption is highest in Vaupes, Huila, Cauca, Quindío, and Arauca, and is slightly higher in Casanare; methamphetamine consumption is highest in Casanare and is moderately high in Boyacá; methadone is most widely used in Quindío, but has slightly high levels of use in Valle del Cauca and Caquetá; opioids are most prevalent in Casanare, followed by Sucre; LSD is most prevalent in Bogotá, but has high levels of use in Caldas, Risaralda, Quindío, and Nariño; mushrooms have their highest consumption in Boyacá and have moderately high uses in Quindío, Risaralda, Bogotá, Cauca, and Casanare; Yagé has a higher incidence in Putumayo; cacao sabanero has its highest consumption in Caldas, and has moderate consumption in Cundinamarca, Bogotá, Antioquia, and Quindío; ketamine has the highest consumption in

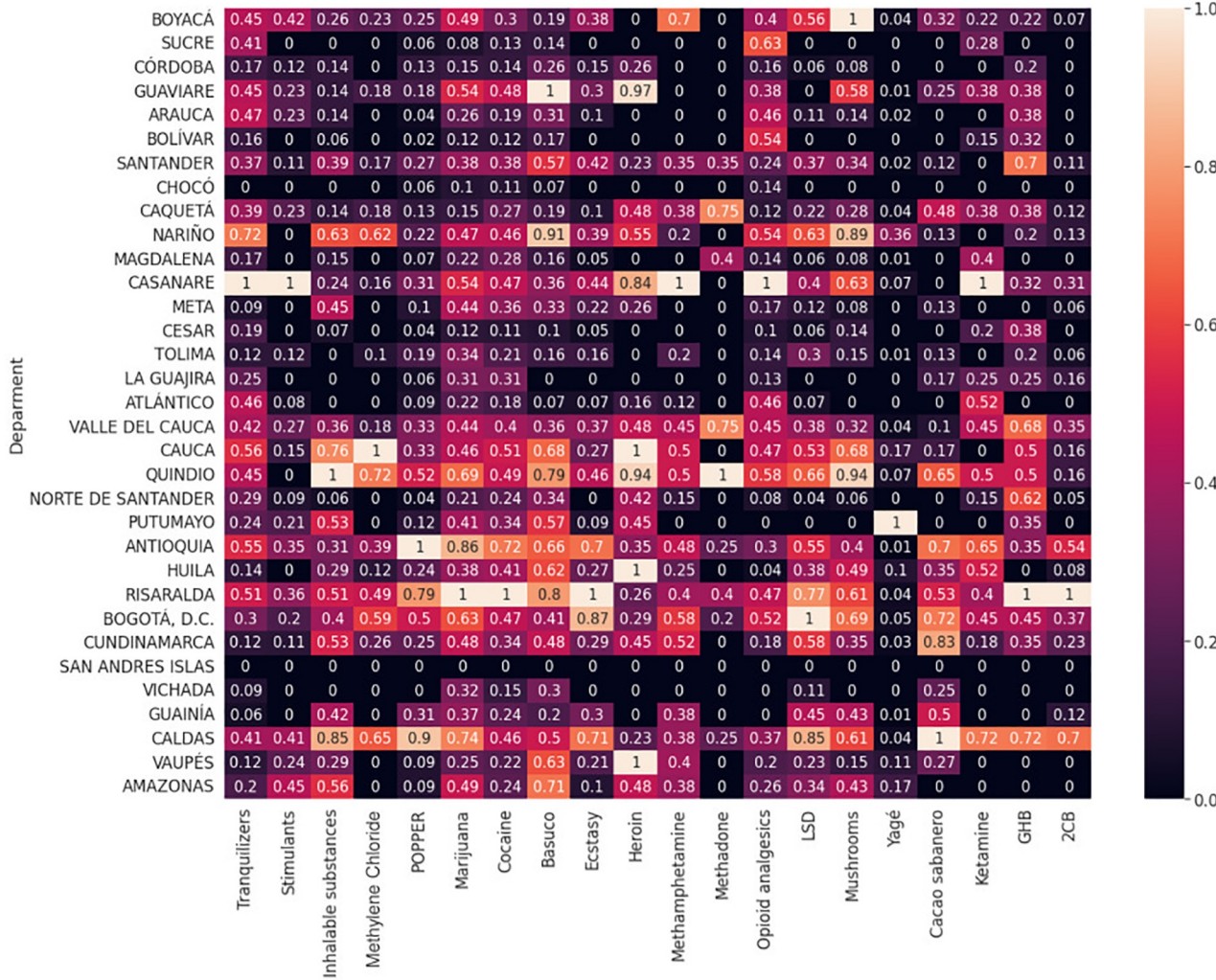

**Fig 8. Consumption of legal drugs by department.** Number represents values scaled on a range of 0 to 1 for psychoactive substance use. Conventions as in Fig 7.

Casanare, followed by Antioquia; and GHB has the highest consumption in Risaralda, followed by Santander, Valle del Cauca, and Norte de Santander. Finally, 2CB has the highest consumption rate in Risaralda, followed by Caldas. Although the consumption pattern of some departments is not mentioned, there is low and moderate consumption for certain drugs in some of them (Fig 8).

## 3.4 Regionalization of clusters

We applied a regionalization method as a grouping technique for imposing a spatial restriction, i.e., the result of a regionalization algorithm contains clusters with geographically coherent areas and coherent data profiles. Our approach uses a spatially constrained hierarchical clustering algorithm, which identified three clusters representing the consumption of PAS in the country (Fig 9). The number of clusters was estimated based on the average silhouette indexes, the total intra-cluster variance, and dendrograms (S2 Fig). Following our results, cluster 0 is comprised of departments such as La Guajira, Cesar, Atlántico, Magdalena, Norte de Santander, Bolivar, Sucre, and Cordoba, all of them located in the Northern region of the country; cluster 1 is comprised of Antioquia, Santander, Boyacá, Caldas, Risaralda, and Quindío; and cluster 2 is integrated by the remaining departments (Fig 9). When testing geographical coherence, which is the measure that assesses the "compactness" of a given shape, our results indicate that the clusters derived using the regionalization model represent moderately compact regions. In addition, the feature coherence (i.e., goodness-of-fit) test using different metrics showed that our 3-cluster regionalization structure properly fits the data (S1 Table).

## 4. Discussion

In this study, we propose and test a Deep Neural Network-based Clustering-oriented Embedding algorithm (i.e., a ML-based model) for identifying psychoactive substance (PAS) use and abuse patterns in Colombia. This model allows the automatic extraction of features from the input data (such as sex, age, socioeconomic status, and housing type) to determine whether an individual has consumed PAS. It then creates clusters in the new data space generated during the learning process, following the methods outlined in [56, 57]. After the training process, a latent feature space (LFS) is generated, and the results are subsequently analysed.

We have identified clearly marked clusters where the prevalence of individuals who use or do not use PAS is notable. Additionally, we found that region, sex, housing type, socioeconomic strata, age, and whether individuals contribute to household finances have a statistically significant impact on the clustering structure. These findings are consistent with previous studies aimed at identifying PAS consumption patterns [19, 79, 80]. Interestingly, when comparing the CAE-DEC model proposed in this study and the CAE-Spectral model using different metrics (i.e., Silhouette statistic, which measures the internal density of each cluster and the distance that separates them from each other, the Calinski-Harabasz index and the Davies-Bouldin index [DBI]), we found that our model performs better (Silhouette: 0.62 vs. 0.786; Calinski-Harabasz: 22468.26 vs. 775992.45; DBI: 0.2898 vs. 0.63; S2 Table).

Based on our findings, individuals more likely to consume PAS are grouped in cluster 2, while cluster 1 consisted of individuals who did not consume PAS (Table 2). Not surprisingly, a significant proportion of females characterizes cluster 1. In addition, most individuals belong to socioeconomic strata 1, are 40 years old or older, and do not contribute economically to support their household. In contrast, cluster 2 is characterized by a higher proportion of males aged between 20 and 40 in socioeconomical strata 1 and 2, who do not contribute to the household finances (Table 2). Finally, cluster 0 is characterized by a small proportion of males, a

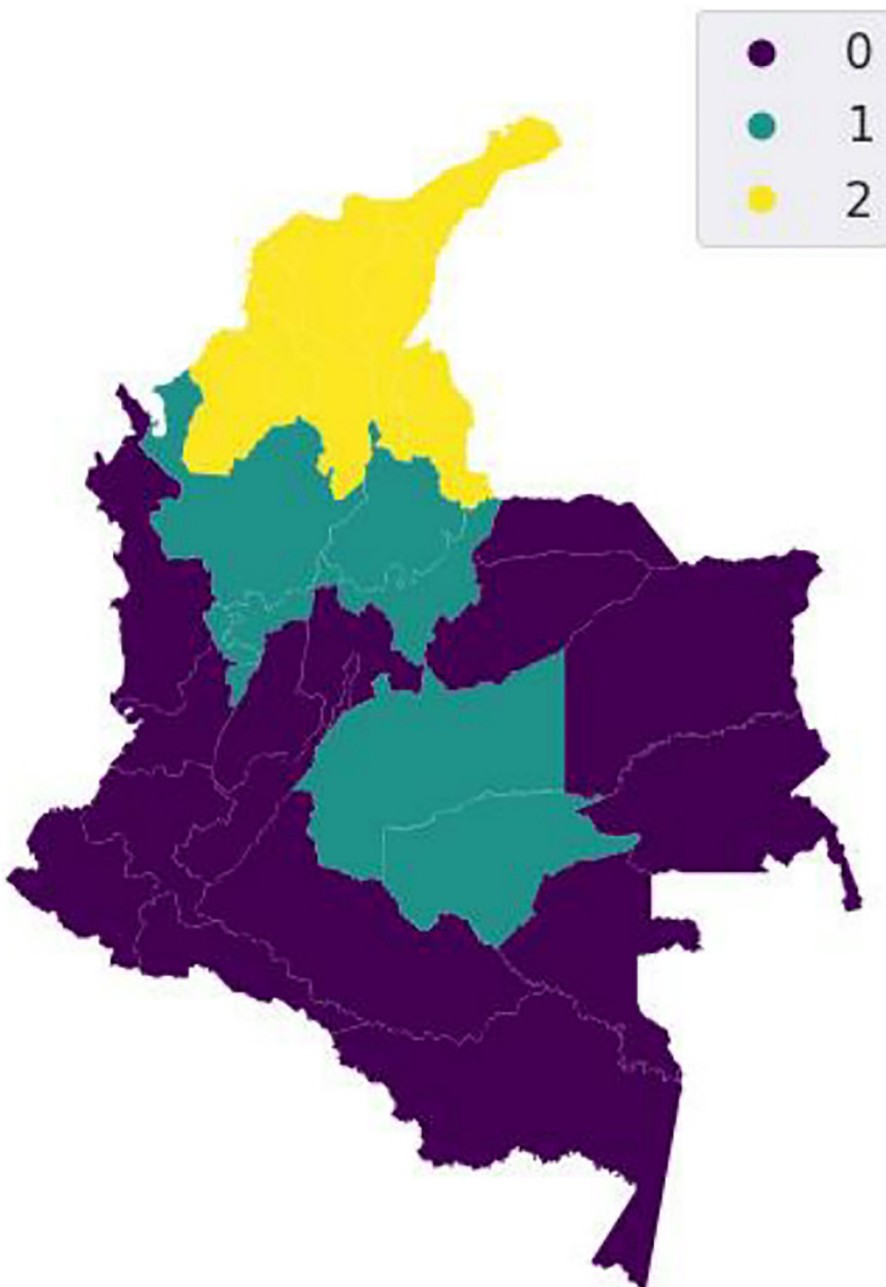

**Fig 9. Cluster map of drug use after regionalization.** Republished from [64] under a CC BY license, with permission from [ArcGIS Hub], original copyright [2016].

higher proportion of individuals in strata 3, 4, 5, and 6, and individuals are more likely to contribute to the household economy (Table 2).

At the level of spatial statistics, we identified that legal drugs such as alcohol have a high prevalence in all regions of Colombia, with a slight tendency to more consumption in coastal areas (Fig 7). In our country, the coastal areas are often popular tourist destinations, and many tourists come to these areas looking for a relaxing experience, which can increase alcohol

consumption. Coastal areas typically have warmer temperatures and more sunshine, increasing thirst and making people more likely to consume beverage. Additionally, bars, clubs, and restaurants serve alcoholic beverage due to the high demand from tourists and locals [81, 82]. Another characteristic of this area is the fishing and maritime culture. This culture is often associated with hard work and long working hours, and alcohol may be seen as a way to relax and unwind after a tough day at the sea [83]. Finally, this region has 69% urban and 31% rural zones [59]. The level of development, as measured by gross domestic product (GDP), is the third region with significant economic development in the country [84] (S3 Table). Interestingly, the consumption of illegal drugs is lower in the Northern region than in other regions of the country. However, there is a more representative consumption of non-prescription tranquilizers, opioids, ketamine, GHB, and heroin. In particular, the Atlántico department has the highest consumption proportion within this region (Fig 8).

Tobacco consumption is present in all regions, with a higher proportion in the Central region (Eje Cafetero–Antioquia), where climate conditions resemble temperate weather. Also, this region has a diverse consumption pattern, where drugs such as marijuana, popper, cocaine, ecstasy, inhalants, methadone, heroin, LSD, GHB, 2CB, and mushrooms prevail. This region has Colombia's largest cities (i.e., Bogotá and Medellin); Bogotá has the highest population density and is a hub for drug trafficking routes, while Medellín has an unfortunate history of drug cartels and gang violence. Ultimately, this region is comprised of 79% urban areas, and the most developed cities in the country are located there [59, 84] (S3 Table).

Energy drinks are more frequently used in the Central-Eastern region, characterized by a continental climate surrounded by flat territory. Our results are in line with the scientific literature suggesting that the location of regions within countries is directly associated with the consumption of PAS [26, 85–87]. The consumption of heroin, basuco, non-prescription tranquilizers, stimulants, methamphetamines, opioids, and ketamine characterizes this region. This zone is the second most developed region in the country, and 71% of urban areas [59, 84], (S3 Table).

Our findings also show that the Southern region is more likely to consume illegal drugs, including basuco, heroin, and Yagé (Fig 8). One of the main reasons for this result is that, unfortunately, this region has favourable environmental characteristics (i.e., majority rainforest) for their consumption and production, being the second largest illegal drug-producing region in Colombia [78]. Furthermore, this region has the highest percentage of rurality (55%) compared to the other regions, and its level of development is low as measured by the GDP [59, 84] (S3 Table).

In the Western region (Pacific), also known as the Pacific region, consumption mostly mainly includes of Methylene Chloride, GHB, heroin, opioids, and methamphetamines. This region (Pacific) is mainly characterized known for its geographical isolation, poverty, and ongoing conflict, which have contributed to the growth of drug production and trafficking in the area. Poverty is one of the main factors driving drug production in the Pacific region, which has led many people to turn to drug cultivation and trafficking for survival. Additionally, the region's rugged terrain and limited infrastructure have made it difficult for the Colombian government to establish a strong presence, allowing drug traffickers to operate with relative impunity [88]. This region has a similar percentage of urban (53%) and rural (47%) populations than the Southern region and ranks second among the regions with the lowest levels of development (S3 Table).

In the Western region, also known as the Pacific region, consumption mainly includes methylene chloride, GHB, heroin, opioids, and methamphetamines. This region is mainly characterized for its geographical isolation, poverty, and ongoing conflict, which have contributed to the growth of drug production and trafficking in the area. Poverty is one of the main

factors driving drug production in the Pacific region, which has led many people to turn to drug cultivation and trafficking for survival. Additionally, the region's rugged terrain and limited infrastructure have made it difficult for the Colombian government to establish a strong presence, allowing drug traffickers to operate with relative impunity [88]. This region has a similar percentage of urban (53%) and rural (47%) populations than the Southern region, and ranks second among the regions with the lowest levels of development (S3 Table) [88]. This region has a similar percentage of urban (53%) and rural (47%) populations than the Southern region. On the other hand, this region ranks second among the regions with the lowest levels of development (S3 Table).

## Conclusion

In summary, the proposed CAE-DEC model simultaneously integrates a feature extraction process within the clustering design, prioritizing features that improve the separation between groups, thus avoiding the manual extraction of features, which is a frequent process in traditional models. Additionally, a geospatial component is sequentially included to expand the resulting insights by considering geographic constraints. Currently, these types of architectures are scarce in understanding mental health problems. As part of future work, the architecture of the proposed model could be improved to integrate the automatic extraction of features while optimizing a geospatial loss. Following our experience with the proposed CAE-DEC in PAS consumption, the application of this model to other mental health problems, such as suicide, depression, and domestic violence, among other pathologies, could be explored. Based on these results, effective interventions and/or government policies to prevent and/or mitigate their impact could be promoted and evaluated, for example, by developing regional interventions based on the types of drugs most prevalent in the area and the cultural and socio-economic characteristics. This can include education, treatment, and harm reduction programs. Also, this information can be used to develop public health campaigns to raise awareness about the risks of drug use and reduce their negative impact. Furthermore, this information can be used to crack down on drug trafficking and distribution networks. On the other hand, this information can be used to alert healthcare providers and regulatory bodies to take appropriate action to prevent their use and discover new drugs.

## Supporting information

**S1 Table. Feature coherence measurements.**
(DOCX)

**S2 Table. Model comparison.**
(DOCX)

**S3 Table. Characteristics of the level of development, urbanity, rurality, and drug production in the regions of Colombia.**
(DOCX)

**S1 Fig. Location map.** Republished from [65] under a CC BY license, with permission from [ArcGIS Hub], original copyright [2016].
(DOCX)

**S2 Fig. The optimal number of clusters using dendrogram.**
(DOCX)

**S3 Fig. Maps of Local Indicators of Spatial Association (LISA).** Republished from [65] under a CC BY license, with permission from [ArcGIS Hub], original copyright [2016]. (DOCX)

## Acknowledgments

K.P. is a doctoral student at Universidad del Norte, Barranquilla, Colombia, and received a Ph. D. scholarship from this institution. Some of this work is to be presented to the Ph.D. program in partial fulfillment of the requirements for the Ph.D. degree.

## Author Contributions

**Conceptualization:** Kevin Palomino.

**Data curation:** Kevin Palomino.

**Formal analysis:** Kevin Palomino.

**Investigation:** Kevin Palomino, Carmen R. Berdugo.

**Methodology:** Kevin Palomino.

**Project administration:** Carmen R. Berdugo.

**Supervision:** Carmen R. Berdugo.

**Visualization:** Kevin Palomino, Jorge I. Vélez.

**Writing – original draft:** Kevin Palomino.

**Writing – review & editing:** Carmen R. Berdugo, Jorge I. Vélez.

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
