## [Decision Letter · Decision Letter 0]

16 Feb 2023

PONE-D-22-28262Leading Consumption Patterns of Psychoactive Substances in Colombia: A Deep Neural Network-based Clustering-oriented Embedding ApproachPLOS ONE

Dear Dr. Palomino,

Thank you for submitting your manuscript to PLOS ONE. After careful consideration, we feel that it has merit but does not fully meet PLOS ONE’s publication criteria as it currently stands. Therefore, we invite you to submit a revised version of the manuscript that addresses the points raised during the review process.

 Also take note of the comments in the attached file.

We look forward to receiving your revised manuscript.

Kind regards,

Vinícius Silva Belo

Academic Editor

PLOS ONE

Journal Requirements:

2. Please confirm that all data sources you used were publicly available and anonymized. Furthermore, please clarify how the data were accessed for the purpose of this study.

3. In the ethics statement in the manuscript and in the online submission form, please provide additional information about the patient records/samples used in your retrospective study. Specifically, please ensure that you have discussed whether all data/samples were fully anonymized before you accessed them and/or whether the IRB or ethics committee waived the requirement for informed consent. If patients provided informed written consent to have data/samples from their medical records used in research, please include this information.

"K.P. is a doctoral student at Universidad del Norte, Barranquilla, Colombia, and received a Ph.D. scholarship from this institution. Some of this work is to be presented to the Ph.D. program in partial fulfillment of the requirements for the Ph.D. degree. The sponsor of the study has no role in study design; in the collection, analysis, and interpretation of data; in the writing of the report; and in the decision to submit the article for publication."

6. We note that Figures 1, 5 9 and S2 in your submission contain map images which may be copyrighted. All PLOS content is published under the Creative Commons Attribution License (CC BY 4.0), which means that the manuscript, images, and Supporting Information files will be freely available online, and any third party is permitted to access, download, copy, distribute, and use these materials in any way, even commercially, with proper attribution. For these reasons, we cannot publish previously copyrighted maps or satellite images created using proprietary data, such as Google software (Google Maps, Street View, and Earth). For more information, see our copyright guidelines: http://journals.plos.org/plosone/s/licenses-and-copyright.

a. You may seek permission from the original copyright holder of Figures 1, 5, 9 and S2 to publish the content specifically under the CC BY 4.0 license.  

Reviewers' comments:

Reviewer's Responses to Questions

**Comments to the Author**

1. Is the manuscript technically sound, and do the data support the conclusions?

Reviewer #1: Partly

Reviewer #2: Partly

2. Has the statistical analysis been performed appropriately and rigorously? 

Reviewer #1: N/A

Reviewer #2: N/A

3. Have the authors made all data underlying the findings in their manuscript fully available?

Reviewer #1: No

Reviewer #2: No

4. Is the manuscript presented in an intelligible fashion and written in standard English?

Reviewer #1: Yes

Reviewer #2: No

5. Review Comments to the Author

Reviewer #1: The manuscript was well constructed. However, the methodology seems very technical, with no explanation of what data were used, how and where the analyzes were carried out. I suggest making the methodology clearer, for example in CAE not only citing authors who used the technique, but making it clear why it was chosen and not another deep learning technique. In the results, improve the identification of tables and figures. Include subtitles to make files more understandable. In the discussion, explore further the impact of the results and assess the environmental issue of these sites. There was a stratification of regions and consumption profile, but there is no information about the location. The area is urban or rural, what level of development. Perhaps these and other local (environmental) aspects may be related and/or favor the consumption of these substances.

Reviewer #2: 1-abstract should rewrite and included technical approach more.

2-page 5 "Feature selection approaches for clustering can be split into filter, wrapper, embedded,"

is it this methods just for clustering !!?

3-in introduction please give problem, challenges clearly.

4-introduction is written separately.

5-in introduction did not cite many reverences together" IDEC (Guo et al., 2017), DEPICT (Dizaji et al., 2017), DBC (F.

14 Li et al., 2017), DualAAE (Ge et al., 2020), VAED (Lim et al., 2020), and DNC (B. Li et al., 2021) "

6- what is difference between convolutional auto-encoder (CAE) and stacked space auto encoder?

7- did not compare other previous works.

8- your problem is classification or clustring?

9-please give your performance metrics

6. PLOS authors have the option to publish the peer review history of their article (what does this mean?). If published, this will include your full peer review and any attached files.

Reviewer #1: No

Reviewer #2: No

---

## [Author Response · Author response to Decision Letter 0]

7 Jun 2023

Dr. Vinicius Silva Belo

RE: Revised Manuscript # PONE-D-22-28262

Dear Dr. Silva, 

Thank you for the opportunity to submit a revised version of our manuscript, "Leading Consumption Patterns of Psychoactive Substances in Colombia: A Deep Neural Network-based Clustering-oriented Embedding Approach,” which now includes responses to the concerns, inquiries, comments, and suggestions raised by two anonymous reviewers. Please find below our response (in blue). 

We very much appreciate all your efforts as Editor-in-Chief and the detailed and extraordinary revision this manuscript had. We enjoyed the thorough review of our manuscript, and it was a pleasure to respond to the reviewers. Please, Dr. Silva, allows us to mention that this exercise of the fair and exigent peer-review process is disappearing, and only good journals like PLOS ONE keep it. We hope that our new version might be suitable for publication in PLOS ONE.

Thank you very much for your time and consideration.

Yours sincerely, 

Kevin R. Palomino, PhD(c) 

Corresponding author 

 

Comments from the Editor

We very much appreciate your comments. We have updated our manuscript according to the journal’s formatting. In particular,

 We changed the figure citation, i.e., from “Figure” to “Fig”.

 We added the Numbered Equation.

 We added the supporting information captions.

 We changed the Cite references into brackets.

2. Please confirm that all data sources you used were publicly available and anonymized. Furthermore, please clarify how the data were accessed for the purpose of this study.

Thank you very much for your comments. The data is fully available and anonymized, and can be retrieved from https://microdatos.dane.gov.co/index.php/catalog/680/data-dictionary

For practicality, we downloaded the data and stored it on the cloud. The data can be retrieved from https://acortar.link/ktgCD9

3. In the ethics statement in the manuscript and in the online submission form, please provide additional information about the patient records/samples used in your retrospective study. Specifically, please ensure that you have discussed whether all data/samples were fully anonymized before you accessed them and/or whether the IRB or ethics committee waived the requirement for informed consent. If patients provided informed written consent to have data/samples from their medical records used in research, please include this information.

Thank you for your comment. As per Colombian regulations, the data were fully anonymized by the Colombian government before we could use them.

K.P. is a doctoral student at Universidad del Norte, Barranquilla, Colombia, and received a Ph.D. scholarship from this institution to cover tuition expenses. However, Universidad del Norte did not provide any additional funds for this study.

Thank you for your comment. The following statement was included in the revised version of the manuscript to address this:

“The funders had no role in study design, data collection and analysis, decision to publish, or preparation of the manuscript”.

As stated previously, we did not receive any salary from organizations.

Done. We have added the following statement in the revised version of the manuscript:

“The authors received no specific funding for this work”.

"K.P. is a doctoral student at Universidad del Norte, Barranquilla, Colombia, and received a Ph.D. scholarship from this institution. Some of this work is to be presented to the Ph.D. program in partial fulfilment of the requirements for the Ph.D. degree. The sponsor of the study has no role in study design; in the collection, analysis, and interpretation of data; in the writing of the report; and in the decision to submit the article for publication."

We appreciate your correction. As per your request, we have removed any funding-related text from the manuscript and clarified that Universidad del Norte provided only tuition expenses for our PhD student.

Thank you!

6. We note that Figures 1, 5 9 and S2 in your submission contain map images which may be copyrighted. All PLOS content is published under the Creative Commons Attribution License (CC BY 4.0), which means that the manuscript, images, and Supporting Information files will be freely available online, and any third party is permitted to access, download, copy, distribute, and use these materials in any way, even commercially, with proper attribution. For these reasons, we cannot publish previously copyrighted maps or satellite images created using proprietary data, such as Google software (Google Maps, Street View, and Earth). For more information, see our copyright guidelines: http://journals.plos.org/plosone/s/licenses-and-copyright.

a. You may seek permission from the original copyright holder of Figures 1, 5, 9 and S2 to publish the content specifically under the CC BY 4.0 license. 

We are very appreciative of your comments. 

Figures containing maps were created using the plot function from GeoPandas (https://geopandas.org/en/stable/), which is an open-source library in Python. This library has a permissive license similar to the BSD 2-Clause License, and it has permissions for distribution, private use, modification, and commercial use (see https://github.com/geopandas/geopandas/blob/main/LICENSE.txt). 

On the other hand, when creating all maps, we used a shape file that contains the location’s geometry. This shapefile was retrieved from https://hub.arcgis.com/datasets/de0e829ddbf743c895ba6dcee1b74fae/about. According to the author, this file can be freely accessible and used, which indicates that neither a license nor permission is needed to use it (i.e., https://hub.arcgis.com/datasets/de0e829ddbf743c895ba6dcee1b74fae/about). 

Kindly see our response to Comment #6. As mentioned above, a license to use the maps is not required.

We very much appreciate your supporting comments. We have updated our manuscript according to the journal’s formatting. 

 We added the Supporting Information captions in the manuscript after references.

 We separated the supplementary materials into four (4) files: S1_Table, S2_Table2, S3_Fig, and S4_Fig.

 We changed the Supporting Information citation on the manuscript i.e. “Table 1S, supplementary material” to “S1 Table”.

 

Comments from Reviewer #1

 The manuscript was well constructed. However, the methodology seems very technical, with no explanation of what data were used, how and where the analyzes were carried out. I suggest making the methodology clearer, for example in CAE not only citing authors who used the technique but making it clear why it was chosen and not another deep learning technique. 

We very much appreciate the reviewer’s comments.

We used data from the Colombian National Survey on Psychoactive Substance Consumption. Data were extracted from https://microdatos.dane.gov.co/index.php/catalog/680/data-dictionary. Analyses were performed using Python’s libraries GeoPandas (https://geopandas.org/en/stable/), TensorFlow (https://www.tensorflow.org/), PyTorch (https://pytorch.org/) and scikit-learn (https://scikit-learn.org/stable/), among others. All notebooks and code written for processing and analyzing the data are available from first author under reasonable request.

Following the reviewer’s suggestion, we have modified the methodology in the revised version of the manuscript. We added a subtitle in the methodology denominated “Framework evaluation.”

Now, the text reads:

“2.8 Framework Evaluation.

We trained the CAE-DEC method using data retrieved from the National Survey of Psychoactive Substance Consumption (DANE-DIMPE-ENCSPA-2019), which contains 49,600 observations. A second database with PAS production figures, was used in the spatial analysis stage to correlate the PSA consumption and production. In order to evaluate the framework, we compared our CAE-DEC approach with other approaches, including CAE, and Principal Component Analysis integrated with clustering (PCA-DEC). For evaluation and comparison purposes, we use the Calinski-Harabasz [76], Davies-Bouldin [77], and Silhouette [78] index as intrinsic clustering metrics. In addition, we used the χ^2 statistic to investigate potential associations and differences among the patterns (clusters) identified using our approach.”

On the other hand, we rewrote a paragraph in the methodology to make clear why the model was chosen. Now, the text reads:

“Fig 1 presents the proposed Convolutional Auto-Encoder- Deep Embedded Clustering (CAE-DEC) framework based on the implementation presented by Xie et al. [57]. However, unlike the Xie et al. model, our structure is developed by applying convolutional layers for the deep autoencoder (DA) architecture instead of a linear one to represent high-order interactions in the data. In addition, a spectral clustering-based centroid estimation is proposed to achieve an improved initial centroid calculation. We chose the CAE-DEC framework based on its ability to reduce both the number of model parameters and the dimensionality, while creating clusters simultaneously.”

”

 In the results, improve the identification of tables and figures. Include subtitles to make files more understandable.

Thank you for your comment. As suggested and following the Journal’s formatting for Figures and Table file naming, we have improved the tables and figures captions in the revised version of the manuscript (i.e., from “Palomino-fig-1” to “Fig1”).

 In the Discussion, explore further the impact of the results and assess the environmental issue of these sites. There was a stratification of regions and consumption profile, but there is no information about the location. The area is urban or rural, what level of development. Perhaps these and other local (environmental) aspects may be related and/or favour the consumption of these substances.

Thank you for your suggestion. We improved the discussion section accordingly in the revised version of the manuscript. The relevant text now reads:

“In this study, we propose and test a Deep Neural Network-based Clustering-oriented Embedding algorithm (i.e., a ML-based model) for identifying psychoactive substance (PAS) use and abuse patterns in Colombia. This model allows the automatic extraction of features from the input data (such as sex, age, socioeconomic status, and housing type) to determine whether an individual has consumed PAS. It then creates clusters in the new data space generated during the learning process, following the methods outlined in [57, 59]. After the training process, a latent feature space (LFS) is generated and the results are subsequently analysed. We have identified clearly marked clusters where the prevalence of individuals who use or do not use PAS is notable. Additionally, we found that region, sex, housing type, socioeconomic strata, age, and whether individuals contribute to household finances have a statistically significant impact on the clustering structure. These findings are consistent with previous studies aimed at identifying PAS consumption patterns [19, 80, 81]. Interestingly, when comparing the CAE-DEC model proposed in this study and the CAE-Spectral model using different metrics (i.e., Silhouette statistic, which measures the internal density of each cluster and the distance that separates them from each other, the Calinski-Harabasz index and the Davies-Bouldin index [DBI]), we found that our model performs better (Silhouette: 0.62 vs. 0.786; Calinski-Harabasz: 22468.26 vs. 775992.45; DBI: 0.2898 vs. 0.63; S2 Table, Supplementary Material).

Based on our findings, individuals more likely to consume PAS are grouped in cluster 2, while cluster 1 consisted of individuals who did not consume PAS (Table 1). Not surprisingly, a significant proportion of females characterizes cluster 1. In addition, most individuals belong to socioeconomic strata 1, are 40 years old or older, and do not contribute economically to support their household. In contrast, cluster 2 is characterized by a higher proportion of males aged between 20 and 40 in socioeconomical strata 1 and 2, who do not contribute to the household finances (Table 1). Finally, cluster 0 is characterized by a small proportion of males, a higher proportion of individuals in strata 3, 4, 5, and 6, and individuals are more likely to contribute to the household economy (Table 1).

At the level of spatial statistics, we identified that legal drugs such as alcohol have a high prevalence in all regions of Colombia, with a slight tendency to more consumption in coastal areas (Fig 7). In our country, the coastal areas are often popular tourist destinations, and many tourists come to these areas looking for a relaxing experience, which can increase alcohol consumption. Coastal areas typically have warmer temperatures and more sunshine, increasing thirst and making people more likely to consume alcohol. Additionally, bars, clubs, and restaurants serve alcoholic beverage due to the high demand from tourists and locals [82, 83]. Another characteristic of this area is the fishing and maritime culture. This culture is often associated with hard work and long working hours, and alcohol may be seen as a way to relax and unwind after a tough day at the sea [84]. Finally, this region has 68% urban and 32% rural zones [60]. The level of development, as measured by gross domestic product (GDP), is the third region with significant economic development in the country [85] (S3 Table, Supplementary Material). Interestingly, the consumption of illegal drugs is lower in the Northern region than in other regions of the country. However, there is a more representative consumption of non-prescription tranquilizers, opioids, ketamine, GHB, and heroin. In particular, the Atlántico department has the highest consumption proportion within this region (Fig 8).

Tobacco consumption is present in all regions, with a higher proportion in the Central region, where climate conditions resemble temperate weather. Also, this region has a diverse consumption pattern, where drugs such as marijuana, popper, cocaine, ecstasy, inhalants, methadone, heroin, LSD, GHB, 2CB, and mushrooms prevail. This region has Colombia’s largest cities (i.e., Bogotá and Medellin); Bogotá has the highest population density and is a hub for drug trafficking routes, while Medellín has an unfortunate history of drug cartels and gang violence. Ultimately, this region is comprised of 77% urban areas, and the most developed cities in the country are located there [60, 85] (S3 Table, Supplementary Material). Energy drinks are more frequently used in the Eastern region, characterized by a continental climate surrounded by flat territory. Our results are in line with the scientific literature suggesting that the location of regions within countries is directly associated with the consumption of PAS [26, [86–88]. The consumption of heroin, basuco, non-prescription tranquilizers, stimulants, methamphetamines, opioids, and ketamine characterizes this region. This zone is the second most developed region in the country, and 72% of urban areas [60], [85](S3 Table, Supplementary Material).Our findings also show that the Southern region is more likely to consume illegal drugs, including basuco, heroin, and Yagé (Fig 8). One of the main reasons for this result is that, unfortunately, this region has favourable environmental characteristics (i.e., majority rainforest) for their consumption and production, being the second largest illegal drug-producing region in Colombia [79]. Furthermore, this region has the highest percentage of rurality (52%) compared to the other regions, and its level of development is low as measured by the GDP [60, 85] (S3 Table, Supplementary Material).

In the Western region, also known as the Pacific region, consumption mostly mainly includes of Methylene Chloride (DICK), GHB, heroin, opioids, and methamphetamines. This region (Pacific) is mainly characterized known for its geographical isolation, poverty, and ongoing conflict, which have contributed to the growth of drug production and trafficking in the area. Poverty is one of the main factors driving drug production in the Pacific region, which has led many people to turn to drug cultivation and trafficking for survival. Additionally, the region’s rugged terrain and limited infrastructure have made it difficult for the Colombian government to establish a strong presence, allowing drug traffickers to operate with relative impunity [89]. This region has a similar percentage of urban (53%) and rural (47%) populations than the Southern region, and ranks second among the regions with the lowest levels of development (S3 Table, Supplementary Material). In the Western region, also known as the Pacific region, consumption mainly includes methylene chloride, GHB, heroin, opioids, and methamphetamines. This region is mainly characterized for its geographical isolation, poverty, and ongoing conflict, which have contributed to the growth of drug production and trafficking in the area. Poverty is one of the main factors driving drug production in the Pacific region, which has led many people to turn to drug cultivation and trafficking for survival. Additionally, the region’s rugged terrain and limited infrastructure have made it difficult for the Colombian government to establish a strong presence, allowing drug traffickers to operate with relative impunity [89]. This region has a similar percentage of urban (53%) and rural (47%) populations than the Southern region, and ranks second among the regions with the lowest levels of development (S3 Table, Supplementary Material).[89]. This region has a similar percentage of urban (53%) and rural (47%) populations than the Southern region. On the other hand, this region ranks second among the regions with the lowest levels of development (S3 Table, Supplementary Material).”

 

Comments from Reviewer #2

 Abstract should rewrite and included technical approach more. 

Done.

 page 5 "Feature selection approaches for clustering can be split into filter, wrapper, embedded," are these methods just for clustering!!?

Thank you for your comment.

No, feature selection approaches do not only apply for clustering algorithms. 

Feature selection is the process of selecting a subset of relevant features (variables and/or attributes) to be used in a supervised or unsupervised model and constitutes an important step in Machine Learning as including irrelevant or redundant features in a model can lead to overfitting, decreased model performance, and increased computational complexity. In this sense, feature selection approaches can be useful for classification, clustering, or regression.

 In the Introduction, please give problem, challenges clearly. Introduction is written separately.

Thank you for your input. Following your advice, several changes have been made in the Introduction of the revised version of the manuscript to address this.

 In introduction did not cite many reverences together" IDEC (Guo et al., 2017), DEPICT (Dizaji et al., 2017), DBC (Li et al., 2017), DualAAE (Ge et al., 2020), VAED (Lim et al., 2020), and DNC (B. Li et al., 2021) "

Done.

 What is difference between convolutional auto-encoder (CAE) and stacked space auto encoder?

Both Convolutional Autoencoder (CAE) and Stacked Autoencoder (SAE) are types of autoencoders, a type of neural network architecture that is used for unsupervised learning and data compression.

The main difference between CAE and SAE is the way they handle the input data. A CAE is typically used for processing image data. It uses convolutional layers to extract spatial features from the input image and then uses deconvolutional layers to reconstruct the image. CAEs are well-suited for image data because they can capture the spatial relationships between pixels in an image and can learn to recognize visual patterns and shapes. On the other hand, SAE is typically used for processing structured or unstructured data. It consists of multiple layers of neural networks that encode the input data into a lower-dimensional representation and then decode it back to the original dimensions. SAEs are useful for feature learning and data compression in many different types of data, including text, audio, and structured data. 

In summary, while CAE is focused on processing image data using convolutional layers, SAE can be applied to various types of data and uses multiple layers of neural networks for encoding and decoding.

 Did not compare other previous works.

Thank you for the comments. In the supplementary section, S2 Table shows the comparison of our model with PCA-K-means, and CAE-Spectral. 

Table S2. Performance metrics for different models.

Performance metric CAE-DEC PCA-K-means CAE-Spectral

Calinski-Harabasz 775992.45 128651.83 22468.26

Davies-Bouldin 0.2898 0.567 0.63

Silhouette 0.786 0.6061 0.62

In this sense, S2 Table shows that our CAE-DEC model results in the highest Calinski-Harabasz score, which means that identified clusters are dense and well separated. On the other hand, our model has the smallest Davies-Bouldin score, which indicates that identified clusters groups have a better partition. Regarding the Silhouette index, our model gives the highest value, which implies that clusters are highly dense.

Performance metric interpretation

 Calinski-Harabasz score: A high score indicates that clusters are dense and well separated. 

 Davies-Bouldin score: Lower is better. Lower values indicating better clustering.

 Silhouette index: The best value is 1 and the worst value is -1. Values near 0 indicate overlapping clusters. Negative values generally indicate that a sample has been assigned to the wrong cluster, as a different cluster is more similar. 

 Your problem is classification or clustering?

We appreciate your question.

The problem we are addressing is this study is a clustering problem, as we aim to identify consumption patterns of psychoactive substances (PAS) in the Colombian territory. In particular, we used clustering techniques (i.e., an ensemble model that integrates an autoencoder and cluster method) to find the different patterns of PAS consumptions in Colombian citizens considering the consumers’ location.

 Please give your performance metrics

The performance metrics are stated in the Discussion of the manuscript. The relevant text reads:

“Interestingly, when comparing the CAE-DEC model proposed herein and the CAE-Spectral model using different score metrics (i.e., Silhouette score, which measures the internal density of each cluster and the distance that separates them from each other, the Calinski-Harabasz index and the Davies-Bouldin index [DBI]) showed that the our proposal model performs better (Silhouette: 0.62 vs. 0.786; Calinski-Harabasz: 22468.26 vs. 775992.45; DBI: 0.2898 vs. 0.63) than the CAE-Spectral alone model”

More information about performance metrics can be found in S2 Table of the Supplementary Material. According to our results, the proposed CAE-DEC model shows a well-separated and highly dense cluster, meaning we can define better groups and identify PAS consumer patterns more precisely.

---

## [Decision Letter · Decision Letter 1]

11 Jul 2023

PONE-D-22-28262R1Leading Consumption Patterns of Psychoactive Substances in Colombia: A Deep Neural Network-based Clustering-oriented Embedding ApproachPLOS ONE

Dear Dr. palomino, 

Thank you for submitting your manuscript to PLOS ONE. After careful consideration, we feel that it has merit but does not fully meet PLOS ONE’s publication criteria as it currently stands. Therefore, we invite you to submit a revised version of the manuscript that addresses the points raised during the review process.

Please submit your revised manuscript by Aug 25 2023 11:59PM. If you will need more time than this to complete your revisions, please reply to this message or contact the journal office at plosone@plos.org. Please include the following items when submitting your revised manuscript:A rebuttal letter that responds to each point raised by the academic editor and reviewer(s). You should upload this letter as a separate file labeled 'Response to Reviewers'.A marked-up copy of your manuscript that highlights changes made to the original version. You should upload this as a separate file labeled 'Revised Manuscript with Track Changes'.An unmarked version of your revised paper without tracked changes. You should upload this as a separate file labeled 'Manuscript'.If applicable, we recommend that you deposit your laboratory protocols in protocols.io to enhance the reproducibility of your results. Protocols.io assigns your protocol its own identifier (DOI) so that it can be cited independently in the future. For instructions see: https://journals.plos.org/plosone/s/submission-guidelines#loc-laboratory-protocols. Additionally, PLOS ONE offers an option for publishing peer-reviewed Lab Protocol articles, which describe protocols hosted on protocols.io. Read more information on sharing protocols at https://plos.org/protocols?utm_medium=editorial-email&utm_source=authorletters&utm_campaign=protocols.

We look forward to receiving your revised manuscript.

Kind regards,

Vinícius Silva Belo

Academic Editor

PLOS ONE

Journal Requirements:

**Additional Editor Comments:**

Please review the comments in the attached file.

Reviewers' comments:

Reviewer's Responses to Questions

**Comments to the Author**

1. If the authors have adequately addressed your comments raised in a previous round of review and you feel that this manuscript is now acceptable for publication, you may indicate that here to bypass the “Comments to the Author” section, enter your conflict of interest statement in the “Confidential to Editor” section, and submit your "Accept" recommendation.

Reviewer #1: All comments have been addressed

2. Is the manuscript technically sound, and do the data support the conclusions?

Reviewer #1: Yes

3. Has the statistical analysis been performed appropriately and rigorously? 

Reviewer #1: Yes

4. Have the authors made all data underlying the findings in their manuscript fully available?

Reviewer #1: Yes

5. Is the manuscript presented in an intelligible fashion and written in standard English?

Reviewer #1: Yes

6. Review Comments to the Author

Reviewer #1: (No Response)

7. PLOS authors have the option to publish the peer review history of their article (what does this mean?). If published, this will include your full peer review and any attached files.

Reviewer #1: No

---

## [Author Response · Author response to Decision Letter 1]

20 Jul 2023

1. I suggest including a location map in the methodology section. The map will offer a clear spatial reference, enabling readers to visualize precise locations of study points or mentioned areas. The inclusion of a location map can improve overall clarity and comprehension, making the article more accessible to a broad audience, including non-specialist readers.

We very much appreciate your comments. We have added a locations map in the methodology section. The relevant text now reads:

“Located in South America, the Republic of Colombia is a diverse country with a population of over 50 million people distributed over a territory of 440,831 square miles [60], encompassing jungles, highlands, grasslands, deserts, coasts, and islands, distributed in six regions and 32 departments (states)[61], (see S1 Fig). It is worth noting that, unfortunately, Colombia has been a major producer of illegal drugs for a long time, which has had a significant impact on drug consumption and abuse.”

2. On page 25, Line 7, I suggest improving the following sentence: “Coastal areas are known to have warmer temperatures and higher levels of sunshine, which can contribute to increased thirst and a higher likelihood of alcohol consumption among people.” The increase in temperature contributes to increased beverage consumption, but not necessarily alcoholic beverages.

Thank you for your suggestion. We improved the discussion section accordingly in the revised version of the manuscript. The relevant text now reads:

In our country, the coastal areas are often popular tourist destinations, and many tourists come to these areas looking for a relaxing experience, which can increase alcohol consumption. Coastal areas typically have warmer temperatures and more sunshine, increasing thirst and making people more likely to consume beverage. Additionally, bars, clubs, and restaurants serve alcoholic beverage due to the high demand from tourists and locals.

---

## [Editor Report · Decision Letter 2]

2 Aug 2023

Leading Consumption Patterns of Psychoactive Substances in Colombia: A Deep Neural Network-based Clustering-oriented Embedding Approach

PONE-D-22-28262R2

Dear Dr. Palomino,

We’re pleased to inform you that your manuscript has been judged scientifically suitable for publication and will be formally accepted for publication once it meets all outstanding technical requirements.

Kind regards,

Vinícius Silva Belo

Academic Editor

PLOS ONE
---

## [Editor Report · Acceptance letter]

10 Aug 2023

PONE-D-22-28262R2 

Leading Consumption Patterns of Psychoactive Substances in Colombia: A Deep Neural Network-based Clustering-oriented Embedding Approach 

Dear Dr. Palomino:

I'm pleased to inform you that your manuscript has been deemed suitable for publication in PLOS ONE. Congratulations! Your manuscript is now with our production department. 

Kind regards, 

on behalf of

Dr. Vinícius Silva Belo 

Academic Editor

PLOS ONE